# A Lipid Bodies-Associated Galactosyl Hydrolase Is Involved in Triacylglycerol Biosynthesis and Galactolipid Turnover in the Unicellular Green Alga *Chlamydomonas reinhardtii*

**DOI:** 10.3390/plants10040675

**Published:** 2021-03-31

**Authors:** Xiaosong Gu, Li Cao, Xiaoying Wu, Yanhua Li, Qiang Hu, Danxiang Han

**Affiliations:** 1Center for Microalgal Biotechnology and Biofuels, Institute of Hydrobiology, Chinese Academy of Sciences, Wuhan 430072, China; xiaosonggu@ihb.ac.cn (X.G.); licao@ihb.ac.cn (L.C.); xiaoyingw@163.com (X.W.); yanhuali@ihb.ac.cn (Y.L.); huqiang@ihb.ac.cn (Q.H.); 2College of Life Sciences, University of Chinese Academy of Sciences, Beijing 100049, China; 3Institute for Advanced Study, Shenzhen University, Shenzhen 518060, China; 4Key Laboratory for Algal Biology, Institute of Hydrobiology, Chinese Academy of Sciences, Wuhan 430072, China; 5State Key Laboratory of Freshwater Ecology and Biotechnology, Institute of Hydrobiology, Chinese Academy of Sciences, Wuhan 430072, China

**Keywords:** *Chlamydomonas reinhardtii*, galactosyl hydrolase, galactolipids, triacylglycerol

## Abstract

Monogalactosyldiacylglycerol (MGDG) and digalactosyldiacylglycerol (DGDG) are the main constituent lipids of thylakoid and chloroplast envelop membranes. Many microalgae can accumulate large amounts of triacylglycerols (TAGs) under adverse environmental conditions, which is accompanied by degradation of the photosynthetic membrane lipids. However, the process mediating the conversion from galactolipids to TAG remains largely unknown. In this study, we performed genetic and biochemical analyses of galactosyl hydrolases (CrGH) identified in the proteome of lipid bodies of the green microalga *Chlamydomonas reinhardtii*. The recombinant CrGH was confirmed to possess galactosyl hydrolase activity by using o-nitrophenyl-β-D-galactoside as the substrate, and the Michaelis constant (Km) and Kcat of CrGH were 13.98 μM and 3.62 s^−1^, respectively. Comparative lipidomic analyses showed that the content of MGDG and DGDG increased by 14.42% and 24.88%, respectively, in the *CrGH*-deficient mutant as compared with that of the wild type cc4533 grown under high light stress conditions, and meanwhile, the TAG content decreased by 32.20%. Up-regulation of CrGH at both a gene expression and protein level was observed under high light stress (HL) conditions. In addition, CrGH was detected in multiple subcellular localizations, including the chloroplast envelope, mitochondria, and endoplasmic reticulum membranes. This study uncovered a new paradigm mediated by the multi-localized CrGH for the conversion of the photosynthetic membranes to TAGs.

## 1. Introduction

Microalgal biomass can be utilized as a feedstock for producing biofuels, owing to its high lipid contents [1,2]. Understanding the process of lipid biosynthesis is essential for the engineering of microalgae for improving lipid production. The unicellular green alga *Chlamydomonas reinhardtii* has been used as a model organism for studying lipid metabolism [3,4]. Like many other microalgae, the *C. reinhardtii* cells can accumulate triacylglycerols (TAGs) when subjected to adverse environmental conditions such as high light and nutrient starvation [2,5]. There are two TAG biosynthesis pathways that have been identified in *C. reinhardtii* [6], the acyl-CoA-dependent pathway and acyl-CoA-independent pathway [4,7,8,9]. The acyl-CoA-dependent pathway involves three sequential acylation reactions and one dephosphorylation reaction [10]. Currently, two glycerol-3-phosphate: acyl-CoA acyltransferase encoding genes (i.e., *CrGPATer* and *CrGPATcl*) [11], two lyso-phosphatidic acid: acyl-CoA acyltransferase encoding genes (i.e., *CrLPAAT1* and *CrLPAAT2*) [12,13], and six diacylglycerol: acyl-CoA acyltransferase encoding genes (i.e., *DGAT1* and *DGTT1*- to -*DGTT5*) have been identified in *C. reinhardtii* [4,7,8]. The acyl-CoA-independent pathway in *C. reinhardtii* is mediated by an ortholog of phospholipid: diacylglycerol acyltransferase (PDAT) gene [9].

Galactolipids are the constitutive lipids of the thylakoid membranes of higher plants, as well as microalgae [14]. Monogalactosyldiacylglycerol (MGDG) and digalactosyldiacylglycerol (DGDG) are two major galactolipids, accounting for about 50% and 30% of thylakoid membrane lipids, respectively [15]. Crystallization studies have shown that MGDG and DGDG are the integral components of PSI [16], PSII [17,18,19], and cytochrome *b6f* complex [20,21], which are essential for the assembly and function of the photosynthetic apparatus [22,23,24]. TAG accumulation in microalgae under adverse stress conditions is usually accompanied by degradation of the photosynthetic membrane lipids, particularly for galactolipids [7,9]. Transmission electron microscopic observation revealed that the stacked thylakoid membranes were dramatically degraded when the *C. reinhadtii* cells accumulated TAG under nitrogen deprivation conditions [25]. Lipidomics analysis indicated that the cellular content of MGDG declined by 63%, while TAG increased around 40 times over two days under nitrogen deprivation [9].

Recently, a number of genes involved in galactolipid turnover have been identified in microalgae and plants. For example, disrupting a MGDG-specific lipase PGD1 (plastid galactolipid degradation 1) in *C. reinhardtii* resulted in a 50% decrease in TAG content, which suggested that the fatty acids utilized for TAG biosynthesis were at least partially recycled from MGDG [26]. The PDAT pathway contributed to approximately 25% of the total TAG accumulated in *C. reinhardtii* under nitrogen deprivation conditions [7], and the in vitro acyltransferase assay indicated that the PDAT of *C. reinhardtii* can utilize MGDG as the acyl donor and DAG as the acceptor for producing TAG [9]. Studies on *Arabidopsis thaliana* have suggested that the gene *SENSITIVE TO FREEZING 2* (*AtSFR2*) coded for a galactolipid, galactolipid galactosyltransferase (GGGT), which transfers the galactosyl moieties from MGDG onto a variety of galactolipid receptors to generate oligogalactolipids (i.e., di-, tri- or tetragalactosyldiacylglycerol) and DAG [27]. However, no ortholog of SFR2 has been identified in the unicellular algae, including *C. reinhardtii* [28], suggesting that more unicellular algae-specific genes/pathways involved in the turnover of galactolipids remain to be identified.

In microalgae, the TAGs synthesized under stress conditions are stored in lipid bodies (LBs), which contained a hydrophobic core surrounded by a polar lipid monolayer coated with proteins [29]. Numerous proteins involved in a variety of functions including lipid metabolism have been identified in the lipid bodies-enriched fraction of *C. reinhardtii* through proteomics analysis [25,30,31,32]. For example, functional classification showed that 58 proteins detected in *Chlamydomonas* LBs fraction were involved in lipid metabolism, including GPAT, LPAAT, and PDAT, which accounted for 9% of the total number of LB proteins. Moreover, several proteins putatively involved in deacylation/reacylation, lipid signaling and lipid trafficking (i.e., TGD1, TGD2, and TGD) were also found to be associated with the LB fraction [32]. Interestingly, it was revealed that the DGDG was present as one of the major membrane lipids in the LBs [33], indicating that the galactolipids might be transported to LBs and then converted to the storage lipids in situ. Thus, dissecting the proteomics of LBs may allow us to identify the novel gene responsible for converting galactolipids to TAG.

To this end, data mining was performed on a previously-reported proteome of *Chlamydomonas* LBs [25], leading to the identification of a putative glycosyl hydrolase (GH) with high abundance. In this study, the LB-associated GH encoding gene was cloned and characterized. By using the purified recombinant protein, the galactosyl hydrolase activity of CrGH was confirmed. A lipidomics analysis performed on the C*rGH*-defective mutant provided lines of evidence supporting the in vivo functions of CrGH being involved in TAG biosynthesis and galactolipids turnover.

## 2. Results

### 2.1. Identification of a Glycosyl Hydrolase Gene from C. reinhardtii

A putative glycosyl hydrolase gene (accession number Cre03.g171100.t1.1, designated as *CrGH*) was identified when conducting a BLASTP search in the *C. reinhardtii* genome database by using the reported LB-associated glycosyl hydrolase protein sequence (Chlre V 3.0 Protein ID:169029) as the query. The 1527-bp open reading frame of *CrGH* was amplified (Figure 1a), and the amino acid sequences showed 100% similarity with the LB-associated glycosyl hydrolase. CrGH belongs to the glycosyl hydrolase family 1, with the conserved active sites (TFNEP and ITETG) (Figure 1b,c), and similar to the structure of glycosyl hydrolase from *Oryza sativa* (OsGH, GenBank: AKS44051), it possesses the representative triosephosphate isomerase (TIM) barrel consisting of eight α-helices and eight parallel β-strands.

The purified recombinant CrGH were approximately 98 kDa (Figure 2a), consistent with the predicted molecular size. The galactosyl hydrolase activity of CrGH was measured by using o-nitrophenyl-β-D-galactoside (ONPG) as the substrate. The results showed that the recombinant CrGH possessed galactosyl hydrolase activities, which can release the o-nitrophenol from ONPG, and the enzymatic activities were positively correlated with the concentration of the enzymes (Figure 2b,c). In addition, the enzymatic kinetics of the galactosyl hydrolase assay in vitro showed that the Michaelis constant (Km) of CrGH was 13.98 μM and the Kcat was 3.62 s^−1^ (Figure 2d).

### 2.2. CrGH Is Involved in TAG Synthesis and Galactolipids Turnover

The enzymatic activity of CrGH in vitro suggested that it might be involved in converting galactolipids into DAG and then into TAG in *C*. *reinhardtii*. To test this hypothesis, a candidate *CrGH*-insertion mutant (LMJ.RY0402.088610) was obtained from the genome-wide insertion library of *C. reinhardtii* mutants for in vivo functional characterization. The gene expression analysis results showed that *CrGH* mRNA abundance significantly decreased, by 92%, in mutant08 compared with that of the wild strain cc4533 (Figure 3a), and CrGH was undetectable in mutant 08 under a N-replete condition (Figure 3b). Moreover, the genome-cassette junctions PCR-based characterization of the insertion sites was performed as shown in the schematic diagram (Figure 3c). The sequencing analysis of the resulting PCR products indicated that the insertion sites were located in the intron of *CrGH* genomic DNA locus (Figure 3d). Based on these results, LMJ.RY0402.088610 was identified as the *CrGH*-deficient mutant. As shown in Appendix A, the cellular growth was not affected by deficiency in *CrGH* under the low light (LL), whereas the growth rate was slightly retarded in the mutant08 grown under high light stress (HL) as compared to the wild type.

Lipidomics analysis of TAG showed that there was no significant difference in TAG content between the mutant08 and the wild type cc4533 under the LL, whereas the TAG in mutant08 was considerably reduced by 32.20% at 96 h compared to that of the wild type cc4533 strain under the HL (Figure 4a). Under LL conditions, as compared with that of the wild type, the fatty acid profile of TAG in mutant08 showed a 55.09% and 75.80% decrease in the 16:0 and C18:1 (n9c), while 69.34%, 87.44%, and 1.24-fold increases were observed in the C18:2 (n6t), C18:2 (n6c), and C18:3 (n6), respectively. Under HL conditions, the percentage of two monounsaturated fatty acids, 16:1 and 18:1, was reduced by 64.66% and 88.09% in the mutant08, respectively. Meanwhile, the percentage of the polyunsaturated fatty acids C18:2 (n6t), C18:2 (n6c), and C18:4 increased by 3.47 and 1.31 times, and 20.00%, respectively (Figure 4b).

To test whether CrGH could utilize galactolipids as the substrate in vivo, the cellular contents of the MGDG and DGDG in the wild type cc4533 and mutant08 strains were quantitatively analyzed by ESI/MS. Under the LL conditions, there was no significant difference in the total contents of MGDG and DGDG between the mutant08 and the wild type cc4533 (Figure 4c,d). Under HL conditions, the contents of MGDG and DGDG in mutant08 increased by 14.42% and 24.88% at 96 h, respectively, as compared to that of the wild type cc4533 (Figure 4c,d). The contents of a number of MGDG and DGDG molecules were significantly altered in the mutant08. Under LL conditions, the content of MGDG 18:3–16:0, MGDG 18:3–16:4, DGDG 18:3–16:3, and DGDG 18:3–16:4 increased by 35.73%, 34.10%, 51.89%, and 33.26%, respectively, in the mutant08 as compared to that of the wild type cc4533 (Figure 4e), though no significant change was detected in the contents of the total galactolipids. Under HL conditions, the content of MGDG 18:2–16:3, MGDG 18:2–16:4, MGDG 18:3–16:4, and MGDG 18:3–16:2 was elevated by 31.65%, 20.92%, 21.01%, and 16.73%, respectively. At the same time, the DGDG species increased to a greater extent than the MGDG species, among which DGDG 18:1–16:4, DGDG 18:3–16:2, and DGDG 18:2–16:3 increased by 16.16%, 31.86%, and 28.96% in the mutant08 as compared to that of wild type cc4533 (Figure 4e). On the other hand, no significant change was detected in the contents of the major extraplastidic membrane lipid DGTS and phosphatidylethanolamine (PtdEtn) between the wild type cc4533 and the mutant08 strains under both LL and HL conditions (Appendix A), suggesting that CrGH was mainly involved in galactolipids metabolism rather than other extraplastidic membrane lipids in *C. reinhardtii*. These results taken together indicated that CrGH might be involved in the remodeling of galactolipids for TAG biosynthesis under HL stress conditions, and it was more likely that CrGH preferred to use the galactolipids with the polyunsaturated fatty acid side chains as the substrates.

### 2.3. Responses of CrGH to HL Stresses and its Subcellular Localization

To confirm whether CrGH was involved in high-light stress responses in *C. reinhardtii* cells, the gene expression was analyzed by quantitative real-time PCR. There was no change in the relative mRNA level in the wild type cc400 cells under the LL conditions, while the transcription of *CrGH* was transiently up-regulated to the maximum level that corresponded to 3.75 times of the control at 6 h under HL. After 12 h, the transcripts of *CrGH* were reduced rapidly to the same levels as the control at 24 h, and then subsequently increased by 2.08 times at 72 h (Figure 5a). At the protein level, there was no significant difference between the mutant08 and the wild type cc4533 under the LL. Under the HL, the abundance of CrGH increased slightly at 6 h and 12 h, reaching 1.41 and 1.48 times of the control, respectively. Subsequently, the abundance of CrGH reduced initial level as the control at 24 h, while at 48 h and 72 h it was up-regulated 1.47 and 1.54 times, respectively (Figure 5b).

Bioinformatic analysis with the online prediction tool ChloroP 1.1, SignalP 4.0, TargetP 1.1, WoLF PSORT, and ProtComp initially failed to predict the subcellular localization of CrGH. The targetP 1.1 prediction analysis showed that the scores of cTP and mTP were 0.192 and 0.134 with the reliability class of 2, respectively, which indicated that CrGH might be a dual-targeting protein. The ProtComp integral prediction of protein location showed that the scores of the prediction in extracellular, cytoplasmic, mitochondrial, peroxisomal, and chloroplast were 2.36, 1.73, 1.25, 2.18, and 1.39, respectively, but no significant similarity was indicated by DBSCAN-P test. In addition, the localizations predicted by comparing the query sequence with its homologs with WoLF PSORT were ambiguous (Appendix A). To resolve the subcellular localization of CrGH, the fractions of chloroplast envelope, mitochondria, endoplasmic reticulum, and thylakoid membranes were isolated from the whole-cell lysates of cc400 cells. Immunoblotting analysis by using the antisera against the marker proteins for each subcellular compartment indicated that high-quality subcellular compartments were obtained (Figure 5c). When using these subcellular compartments for analyzing the subcellular localization of CrGH, it was detected in the chloroplast envelope, mitochondria, and endoplasmic reticulum fractions. These results suggested that CrGH was a multi-localization targeting protein in *C. reinhardtii* cells.

## 3. Discussion

TAG and galactolipid metabolism are believed to be closely interactive in microalgae [6,9,34]. As the major components of the photosynthetic membranes, galactolipids are subjected to photo-oxidative stresses under various adverse environmental conditions. Biosynthesis of TAG molecules can sequester excess electrons from the photosynthetic electron transport chain for reducing the oxidative damage to photosystems [2]. A previous study showed that disrupting the galactolipid lipase encoding gene *PGD1* in *C. reinhardtii* resulted in a 50% decrease of the TAG content and chlorotic appearance during nitrogen deprivation. As a result, over-reduction of the photosynthetic electron transport chain caused severe damage to thylakoid membranes and led to the cell death for the mutant [26]. In this study, the results of in vitro and in vivo analyses taken together indicated that CrGH is a galactosyl hydrolase involved in TAG biosynthesis and galactolipids degradation under HL stress. Indeed, it represents a new paradigm in which the most abundant photosynthetic membrane lipids could be converted to DAG through hydrolysis of the galactose residue and then be utilized for TAG biosynthesis. Moreover, nitrogen deprivation is an important stress treatment to induce the TAG accumulation in *Chlamydomonas reinhardtii*. Therefore, we are investigating whether the CrGH is involved in converting galactolipids into the TAG under N-deprivation conditions and other abiotic stress conditions, such as high salinity, and the experiments are underway.

Under HL stress, the cone-like shaped MGDG tends to form an inverted hexagonal phase (HⅡ) [35], which can cause the fusion of photosynthetic membranes, and is thus deleterious for photosynthetic cells. However, the *CrGH* deficient mutant did not show any defect in growth under HL stress (Appendix A). It is noteworthy that only a few MGDG species showed a slight increase in the mutant, while the changes in the DGDG content were more substantial than that of MGDG. In addition, CrGH was detected in both the chloroplast envelope and the extraplastidic compartments, including mitochondria and endoplasmic reticulum. Based on these observations, we speculated that it was most likely that CrGH mainly acted on DGDG, instead of MGDG. DGDG is a type of typical bilayer lipid that has been identified in multiple subcellular compartment membranes, in addition to the chloroplasts of plant cells, including mitochondria and plasma membranes [36]. A previous study showed that DGDG was present on the surface of LBs in *C. reinhardtii* cells [33], suggesting that DGDG can be exported from the chloroplasts to extraplastidic organelles when algal cells accumulate enormous amounts of TAGs. In addition, there are extensive lines of evidence showing that LBs interact with a variety of cell organelles, including the endoplasmic reticulum, mitochondria, chloroplast, and peroxisomes, presumably via the membrane contact sites [28,37,38].

In this study, we failed to detect the formation of galactose when commercial and purified galactolipids of *C. reinhardtii* were used as the substrates to test the enzymatic activity of the recombinant CrGH (data not shown). This could be attributable to the alterations in conformation of the membrane-bound galactolipids in solvents. The MGDG bilayer formation requires the participation of the relative proteins, but after extraction form *C. reinhardtii*, it may form a hexagonal (HⅡ) non-lipid bilayer structure, thus changing the enzyme recognition for impeding the catalytic process of the CrGH in vitro. At present, the related protein homology modeling of CrGH has a lack of valid templates supported by experimental data. Therefore, crystal analysis and molecular docking of CrGH will be investigated in a future study to dissect the catalytic mechanism. In this study, the multi-targeting subcellular localizations of CrGH indicated post-translational modification is likely essential for the authentic activity of CrGH in vivo. Therefore, we will also use the recombinant protein expressed by *S. cerevisiae* INVSC1 for future study. On the other hand, it could not be excluded that the CrGH may function as GGGT in vivo. A recent study confirmed that *SFR2* coding AtGGGT localized at the outer chloroplast envelope membranes of *Arabidopsis* can stabilize the chloroplast membrane structure under freezing stress condition owing to its galactosyltransferase activity, with the formation of oligogalactolipids and DAG by transferring the galactosyl moiety from one MGDG molecule onto the other galactolipid acceptor [27]. In addition, AtSFR2 displayed a specific hydrolytic activity against β-glucosidase in vitro [39]. Although previous studies indicated that *C. reinhardtii* does not possess the ortholog of GGGT [28], it is necessary to conduct thorough in vivo and in vitro investigations to test whether CrGH could function as GGGT.

## 4. Conclusions

An algal LB-associated putative glycosyl hydrolase encoding gene was cloned, and the corresponding recombinant protein showed typical galactosyl hydrolase activity in vitro. The in vivo function of CrGH was found to be involved in TAG biosynthesis and turnover of galactolipids through characterizing an *CrGH* insertion mutant. In addition, the up-regulation of *CrGH* at both a gene expression and protein level was observed under HL conditions. In summary, this study uncovered a novel paradigm mediating the conversion of the photosynthetic membranes to the storage lipids. The physiological functions of CrGH under other abiotic and biotic stress conditions remain to be investigated in the future.

## 5. Materials and Methods

### 5.1. Strains and Growth Conditions

*C. reinhardtii* cc-400 cw15 mt+, cc4533, and *CrGH* deletion mutant were obtained from the Chlamydomonas Resource Center (http://chlamycollection.org/, Minnesota University). The algal cell cultures were grown in Tris-acetate-phosphate (TAP) medium at 23 °C under a continuous illumination of 50 µmol photons/(m^2^·s). For induction of TAG accumulation, the algal cells in the exponential growth phase grown under low light (LL) conditions at 50 µmol photons/(m^2^·s) were harvested by centrifugation at 3000× *g* for 5 min, resuspended in fresh Sueoka’s high salt medium (HSM) at a starting cell density of 2 × 10^6^ cells/mL, and then subjected to the high-light (HL) conditions, corresponding to the illumination of 200 µmol photons/(m^2^·s). Cell aliquots were collected at defined time intervals. All the cell cultures were grown on an orbital shaker (NBS Innova 44R, Eppendorf, Germany) with continuous shaking at 120 rpm.

### 5.2. Bioinformatics Analysis

To identify the gene coding for the glycosyl hydrolase identified in LBs, a BLASTP search of the *C. reinhardtii* genome database (https://phytozome.jgi.doe.gov/pz/portal.html) was performed by using the reported protein sequence (Chlre V 3.0 Protein ID:169029) as the query. The gene annotated as glycosyl hydrolase (*CrGH*, annotation number: Cre03.g171050.t1.1) was obtained. The cDNA of *CrGH* was amplified by PCR from the cDNA library with the primers *CrGH*-F/R (Appendix A). The PCR fragment were then cloned into the pEASY-Blunt simple cloning vector (Transgen, China) for sequencing. The protein transmembrane helices, putative transit peptide, and signal sequences were identified by using TMHMM Server v. 2.0, ChloroP 1.1, and SignalP 4.0, respectively. In addition, the software TargetP 1.1 (http://www.cbs.dtu.dk/services/TargetP-1.1/index.php), WoLF PSORT (https://wolfpsort.hgc.jp/), and ProtComp (http://www.softberry.com/) were used to predict the subcellular localization. The amino acid sequences were aligned by DNAMAN software, and consensus sequences were highlighted using BOXSHAD (https://embnet.vital-it.ch/software/BOX_form.html).

### 5.3. Heterologous Expression of the Recombinant CrGH in Escherichia coli

The full-length CrGH coding sequence was amplified from the pEASY/CrGH plasmid with the primers *CrGH*-*mbp*-*linker*-F/R (Appendix A) and was then constructed into the pMAL-c5x vector (NEB, Ipswich, MA, USA) containing the maltose binding protein (MBP) tag. *Escherichia coli* BL21 (DE3) cells were transformed with the constructed plasmids for protein expression. The single transformation was inoculated to 10 mL of Luria–Bertani (LB) medium with ampicillin (100 µg/mL final concentration) and grown at 37 °C overnight on an orbital shaker at a speed of 220 rpm. Approximately 0.5 mL of the cell culture was added to 50 mL of LB medium and grown at 37 °C for 3 h. When the OD_600_ reached about 2, the cells were cultured at 30 °C for 16 h with the addition of isopropyl-β-D-thiogalactopyranoside (IPTG, 0.4 mM final concentration). The cells were harvested by centrifugation at 5000× *g* for 10 min at 4 °C. Then, the pellets were frozen in liquid N_2_, and stored at −80 °C until use.

Purification of the recombinant protein was performed as instructed by the manufacturer’s instructions (NEBExpress^®^ MBP Fusion and Purification System, NEB, Ipswich, MA, USA). Briefly, the harvested *E. coli* cell pellets were resuspended in 25 mL of the column buffer containing 20 mM Tris-HCl (pH 7.4), 200 mM NaCl, 1 mM EDTA, 1 mM DTT, and 1 × cocktail proteinase inhibitors (Roche, Switzerland). The mixture was sonicated to lyse cells. Lysates were centrifuged at 20,000× *g* for 15 min at 4 °C and the supernatant was mixed well with amylase resin (NEB, Ipswich, MA, USA) overnight with rotation at a low speed. The mixture was loaded onto a 2.5 × 10 cm column, and was washed with 12 column volumes of column buffer. The target protein was eluted by column buffer with 10 mM maltose. The purified protein was checked after separation on SDS-PAGE gel. Protein concentration was determined with a CB-X™ Protein Assay kit (GBioscience, St. Louis, MO, USA).

### 5.4. Antisera Production and Purification

The PCR fragments of *CrGH* were digested with *EcoR* I/*Xho* I, and then ligated with the enzymatic digested pGEX-4T-1 vector (GBiosciences, St. Louis, MO, USA). The induced cell pellets of *E. coli* BL21 (DE3) strains harboring the pGEX-4T-1-*CrGH* constructs were disrupted by sonication to isolate the inclusion bodies, and the recombinant CrGH in the form of inclusion bodies were directly recovered from the SDS-PAGE and were used as the antigen to raise antiserum in rabbits by following the standard protocol (ABclonal Biotechnology Co., Ltd., Wuhan, China). The antigen was prepared with Freund’s adjuvant and then injected into rabbits with three additional boosts. The final purified antigen-specific polyclonal antibodies were used as the primary antibody for immuno-blotting analysis.

### 5.5. In Vitro Enzymatic Assay with the Recombinant CrGH

Galactosyl hydrolase activity of the recombinant CrGH protein was assayed by using a β-Galactosidase kit (Solarbio, Beijing, China) according to the instructions. In a 500 µL reaction, different quantities of the purified protein (20, 100, and 500 µg) were mixed with the substrate of 3 µM o-nitrophenyl-β-D-galactoside (ONPG) in 30 mM magnesium chloride solution buffer (pH 7.3) for incubation at 37 °C for 15 min. Then, the reactions were stopped by addition of 300 µL of 0.4 M sodium carbonate and the o-nitrophenol produced via the hydrolytic reaction was measured according to its optical absorbance at 400 nm. The β-galactosidase from *E. coli* (Sigma-Aldrich, St. Louis, MO, USA) was used as a positive control. For the enzymatic kinetics determination, the different gradient concentrations of o-nitrophenyl-β-D-galactoside (0.15, 0.375, 0.75, 1.5, 2.25, 3, 7.5, 12, 18, 24, and 30 µmol) were incubated with approximately 20 µg of the purified CrGH protein in the reaction system. All the experiments were performed with at least three replicates. Values are presented as means ± SD. The enzyme kinetics analyses were performed using Microsoft Excel for curve-fitting.

In addition, we used both commercial and purified galactolipids of *C. reinhardtii* for the in vitro assay, as mentioned above. The commercial galactolipids included: the MGDG mixture (Avanti Polar Lipids, Alabaster, AL, USA) consisting of 66.8% of MGDG 16:3–18:3, 14.1% of MGDG 18:3–18:3, 12.9% of MGDG 16:3–18:2, 3.2% of MGDG 18:2–18:3, and 3.0% of MGDG 16:1–18:3; and DGDG mixture (Avanti Polar Lipids, Alabaster, Alabama, USA) consisting of 44.5% of DGDG 18:3–18:3, 21.3% of DGDG 16:3–18:3, 10.7% of DGDG 18:2–18:3, 9.7% of DGDG 16:0–18:3, 7.0% of DGDG 16:3–18:2, and 6.9% of DGDG 16:1–18:3. The MGDG and DGDG of *C. reinhardti* were also extracted and purified with thin-chromatography chromatography using chloroform/hexane/tetrahydrofuran/isopropanol/water (50:100:1:80:2, *v*/*v*/*v*/*v*/*v*) as the solvent system. The developed TLC plates were air-dried and sprayed uniformly with solid iodine at room temperature for 10 min. The MGDG and DGDG spots were scraped from the TLC plates and then dissolved in chloroform/methanol (1:1, *v*/*v*) solutions. The solvents were then evaporated under nitrogen stream. Lipid samples were stored at −80°C prior to use.

### 5.6. RNA Extraction and Real-Time Fluorescent Quantitative PCR

Total RNAs were extracted using TransZol Up reagent (TransGen, Beijing, China). The cDNA was generated through reverse transcription using EasyScript One-Step cDNA Removal and cDNA Synthesis SuperMix kit (TransGen, Beijing, China). Quantitative real-time PCR was performed with a LightCycler^®^ 480 System according to the manufacturer’s instructions and using the LightCycler^®^ Multiplex Masters (Roche, Basel, Switzerland) with the primers CrGH-qPCR-F/R (Appendix A). The *α*-*tubulin* gene was used as the internal reference.

### 5.7. Verification of the Mutant Defective in CrGH

A Chlamydomonas mutant (LMJ.RY0402.088610) generated by electroporation of a DNA cassette (CIB1) conferring resistance to paromomycin into Chlamydomonas strain cc4533 [40] was purchased from the Chlamydomonas Resource Center. To verify the inserted mutation defective in *CrGH*, the genomic DNA of LMJ.RY0402.088610 was extracted using an EasyPure^®^ Plant Genomic DNA Kit (TransGen, Beijing, China). Two pairs of primers (primers G1/C1 and primers G2/C2, Appendix A) were utilized to amplify the cassette–genome junction on both sides of each insertion site, as indicated by the flanking sequences [41]. The 25 μL PCR reaction mixtures contained 2.0 μL 2.5 mM dNTPs, 16 μL of nuclease-free water, 0.5 μL of each primer at 10 μM, 0.5 μL of 50 ng/μL *C. reinhardtii* genomic DNA, 0.5 μL *TransStart^®^ FastPfu* DNA polymerase, and 5 μL 5×*TransStart^®^ FastPfu* Buffer (TransGen, Beijing, China). PCR amplification parameters were: 3 min at 95 °C, 35 cycles of 30 s at 95 °C, 30 s at 54 °C, 45 s at 72 °C, followed by a final extension of 5 min at 72 °C. The PCR amplification products were purified for sequencing. The resulting sequences were aligned against the *C. reinhardtii* genome sequence from Phytozome v5.3.

### 5.8. Lipid Extraction and Fatty Acids Profiling

*C. reinhardtii* cells were collected by centrifugation at 3000× *g* for 5 min and the pellets were lyophilized. Total lipids were extracted according to the Bligh and Dyer method [42] and separated on a silica thin layer chromatography (TLC) plate (Merck, Germany) using petroleum ether/ethyl ether/acetic acid (70:30:1, *v*/*v*/*v*) as the solvent system. For the visualization of the separated glycerides, the developed TLC plates were air-dried and sprayed uniformly with solid iodine at room temperature for 10 min. The TAG spots were scraped from the TLC plates and then trans-esterified to the fatty acid methyl esters for gas chromatography-mass spectrometry analysis by using an Agilent 7890B gas chromatograph coupled with 59977A mass spectrometry (GC-MS, Santa Clara, CA, USA), as previously described [9].

### 5.9. Quantitatively Analysis of the Membrane Lipids with Electrospray Ionization Mass Spectrometry (ESI/MS)

Membrane lipids analyses were performed according to the previously described lipidomic method [9] with modifications. Lipidomes analyses were performed on a triple quadrupole MS/MS (Xevo TQ-S, Waters, Milford, MA, USA) with electrospray ionization (ESI) source coupled with an Acquity Ultra-Performance Liquid Chromatography (UPLC) system (Waters, Milford, MA, USA). The extracted lipids were dissolved in 1 mL chloroform/methanol (1:1, *v*/*v*) and then were mixed with internal standards (ISTDs), including MGDG 18:0/18:0, DGDG 18:0/18:0, 1, 2-diacylglyceryl-3-*O*-4′-(*N*, *N*, *N*-trimethyl)-homoserine (DGTS) 16:0/16:0 d9, phosphatidylglycerol (PG) 17:0/20:4, phosphatidylethanolamine (PE) 17:0/14:1, and phosphatidylinositol (PI) 17:0/20:4. The MGDG mixture standard containing MGDG 16:1/18:3, MGDG 16:3/18:2, and MGDG 16:3/18:3; DGDG mixture standard containing DGDG 16:0/18:3, DGDG 16:1/18:3, DGDG 16:3/18:2, and DGDG 16:3/18:3; DGTS 16:0/16:0, PG 16:0/18:1, PG 18:0/18:1, PE 18:0/18:1, PI 18:1/18:1, and sulfoquinovosyldiacylglycerol (SQDG) 16:0/18:3 were used as external standards (ETSDs) for the corresponding classes of membrane lipids. All lipid standards were purchased from Avanti Polar Lipids Ltd. (Alabaster, AL, USA) except for MGDG 18:0/18:0 and DGDG 18:0/18:0, which were purchased from Matreya LLC (State College PA, USA). For absolute quantification, ESTDs were titrated relative to a constant amount of ISTD to establish the correlation between the ratio of the analyte signal to the ISTD signal and the ratio of their concentrations. In this study, multiple reaction monitoring (MRM) was employed for quantitative analysis. All experiments were repeated three times on different biological samples, each of which contained three technical replicates.

### 5.10. Preparation of Subcellular Compartments

To obtain cellular organelles extracts of *C. reinhardtii*, four liters of cc400 cell cultures during the logarithmic period was harvested by centrifugation at 3000× *g* for 5 min at 4 °C and washed with ice-cold PBS buffer. Cell pellets were used for organelle fraction extraction. Mitochondria and ER membranes were isolated as previously described [43,44], respectively. Chloroplast envelope membranes and thylakoid membranes were isolated as previously described [45,46,47] with some modifications. The pellets were resuspended in breaking buffer containing 0.3 M sorbitol and 50 mM HEPES (pH = 7.5), and 2 mM EDTANa_2_ (pH = 8.0), 1 mM MgCl_2_, and 1 × cocktail protease inhibitors to a concentration of 10^8^ cells/mL, and the diluted cells were rapidly drawn into the syringe with a 27-gauge needle and passed through the needle at a flow rate of 0.25 mL/s. The cell homogenates were centrifuged at 1000× *g* for 5 min to pellet chloroplasts and unbroken cells. Pellets were resuspended in chloroplast lysis buffer containing 50 mM Tris-HCl (pH = 7.5) and 2 mM MgCl_2_, and homogenized with a Potter–Elvehjem tissue grinder. The homogenates were layered onto a sucrose gradient containing 0.9 M/0.6 M sucrose in the chloroplast lysis buffer and centrifuged at 10,000× *g* for 60 min at 4 °C. The membrane fractions at the interface of 0.9 M/0.6 M sucrose were collected as the chloroplast envelope membranes. The pellet was further resuspended in 1.8 M sucrose in buffer A containing 50 mM Tris-HCl (pH 7.5) and 10 mM EDTANa_2_ (pH 8.0), and homogenized with a Potter–Elvehjem tissue grinder. The homogenates were layered onto a sucrose gradient containing 1.3 M/0.6 M sucrose in buffer A and centrifuged at 100,000× *g* for 60 min. Thylakoid membranes floating on the 1.3 M sucrose were collected.

### 5.11. Immunoblotting Analysis

The isolated organelle fractions were resuspended in a mixture of 60 µL of buffer B containing 0.1 M dithiothreitol and 0.1 M Na_2_CO_3_, and 40 µL of buffer C containing 30% (*w*/*v*) sucrose and 5% (*w*/*v*) SDS, and vortexed at 3000 rpm for 30 min. Insoluble proteins were removed by centrifugation at 12,000× *g* for 10 min. The protein concentration was measured using a CB-X protein assay kit (GBiosciences, St. Louis, MO, USA). Proteins were loaded onto 12% SDS-PAGE gels and transferred to a nitrocellulose membrane. Antibodies against CrGH (prepared by ABclonal Biotechnology Co., Ltd., Wuhan, China), Toc34 (AS07238, Agrisera), Bip (AS09481, Agrisera), Aoxi (AS06152, Agrisera), PsbA/D1 (AS05084, Agrisera) and α-Tubulin (AS10680, Agrisera) were used at 1:1000, 1:10000, 1:2000, 1:10000, 1:10000, and 1:1000 dilutions, respectively. Secondary anti-rabbit antibodies (1706515, Bio-Rad) and anti-mouse antibodies (1706516, Bio-Rad) were both used at 1:1000 dilution. Immunoblotting signals were visualized with an enhanced chemiluminescence (ECL) assay kit (Vazyme, China) according to the manufacturer’s protocol.

## Figures and Tables

**Figure 1 plants-10-00675-f001:**
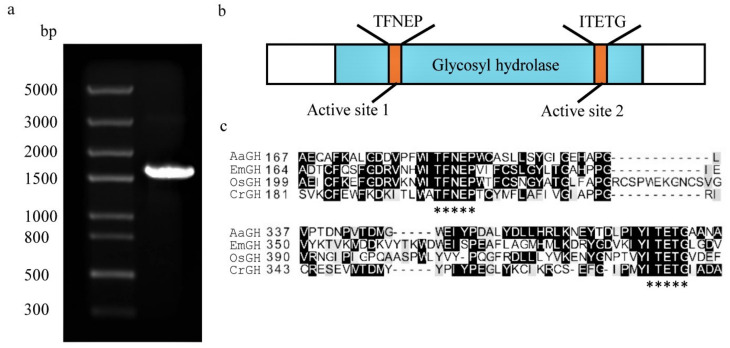
Cloning of the galactosyl hydrolase gene from *Chlamydomonas reinhardtii* and bioinformatics analysis. (**a**). The PCR amplification of *CrGH* (~1527 bp). (**b**). Functional domains of the putative CrGH, which consists of active sites (turquoise boxes) and glycosyl hydrolase domain (orange box). (**c**). Multiple protein sequence alignment of glycosyl hydrolases of *Alicyclobacillus acidiphilus* (AaGH, accession number: WP_067621817.1), *Exiguobacterium marinum* (EmGH, accession number: WP_026824377.1), *Oryza sativa* (OsGH, accession number: 3PTQ_B), and *C. reinhardtii* (CrGH, GenBank: Cre03.g171050.t1.1). The sites of the active site peptide motifs (TFNEP and ITETG) are indicated (*).

**Figure 2 plants-10-00675-f002:**
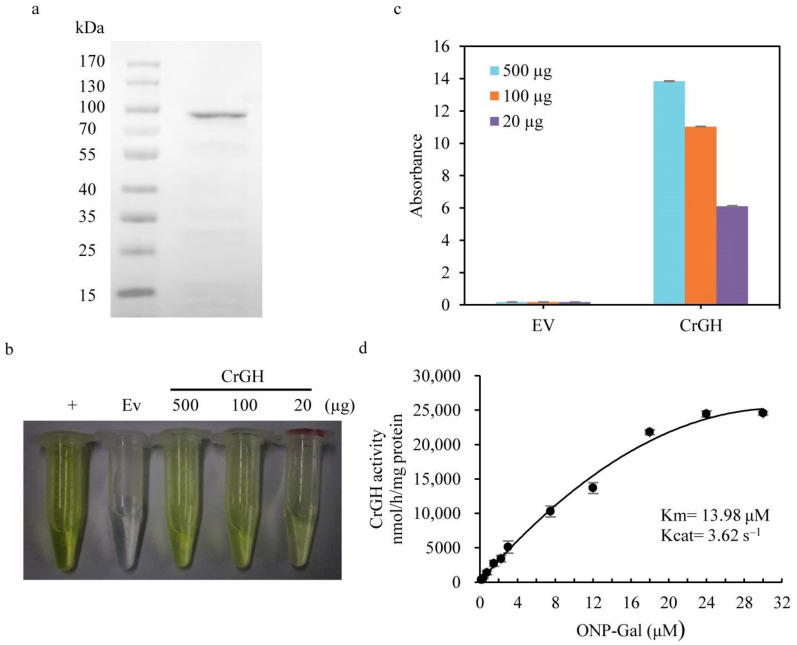
In vitro assay of the galactosyl hydrolase activity for the recombinant CrGH. (**a**). Expression of the recombinant CrGH in *E. coli*. *(***b**). Enzymatic assay for CrGH with the substrate of ONP-β-D-galactopyranoside. EV, empty vector. The β- galactosidase from *E. coli* was used in the enzymatic reactions as a positive control. (**c**). Quantitation of the produced o-nitrophenol in enzymatic assays with various quantities of CrGH. (**d**). Kinetic determination of galactosyl hydrolase of the recombinant CrGH.

**Figure 3 plants-10-00675-f003:**
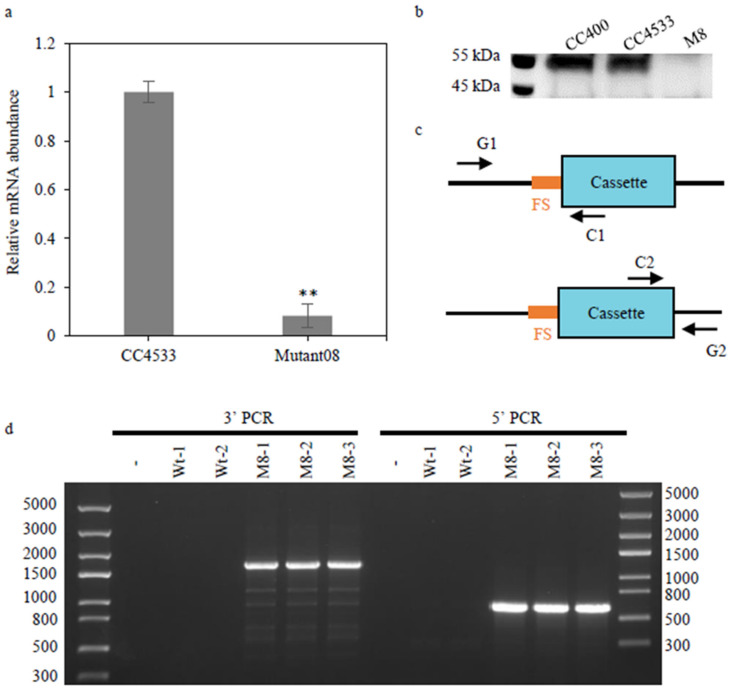
Verification of the insertion in *CrGH*. (**a**). Relative level of the *CrGH* mRNA transcript by real-time fluorescent quantitative PCR. The abundance of mRNA was normalized to the mRNA transcript level in wild type cc4533 strain, which was set as 1. Data are presented as average ± SD (n = 3) * *p* < 0.05, ** *p <* 0.01. (**b**). Immunoblotting analysis of the protein levels of CrGH in the wild type (cc4533 and cc400) and mutant08 strains. Twenty micrograms of the whole-cell proteins were loaded in each lane. (**c**). Schematic diagram of genome-cassette junction amplification. The flanking DNA sequences (FS) were amplified with PCR primers binding to the cassette and genome. (**d**). Amplification of genome-cassette with the locus disruption. The resulting PCR product was sequenced for identifying the site of each insertion.

**Figure 4 plants-10-00675-f004:**
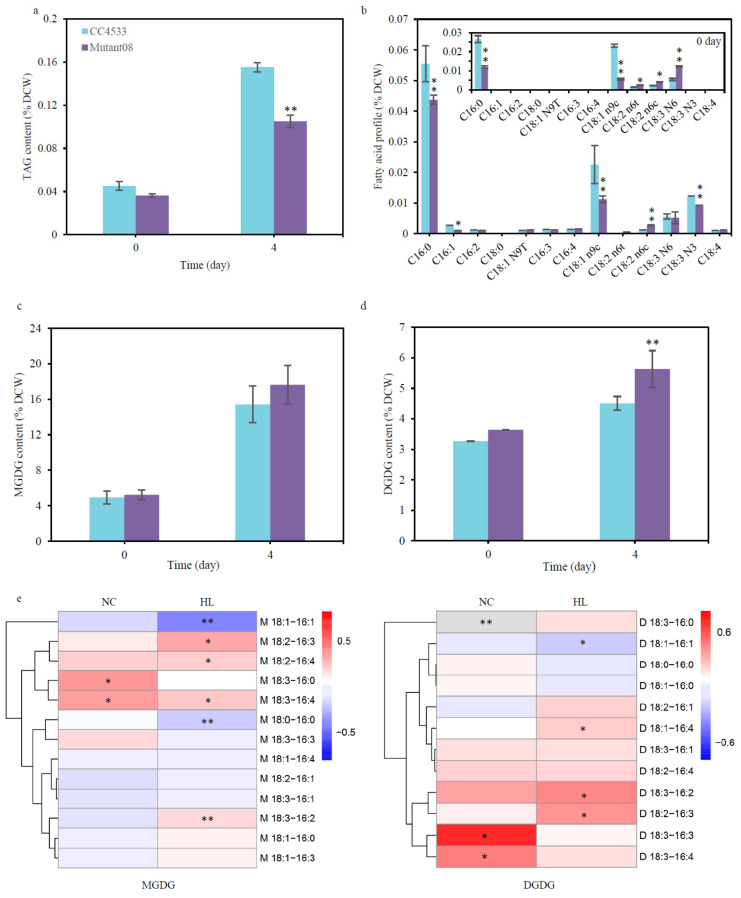
Insertional mutation of *CrGH* suppressed TAG accumulation and resulted in increases in the contents of galactolipids. (**a**). TAG contents of the wild type cc4533 and mutant08 grown under low light (LL) (day 0) and high light stress (HL) (day 4). (**b**). Fatty acid profile of the TAG. (**c**). Monogalactosyldiacylglycerol content. (**d**). Digalactosyldiacylglycerol content of the wild type cc4533 and mutant08 grown under LL (day 0) and HL (day 4). (**e**). Heatmap of the fold-changes in the galactolipids content in mutant08 as compared with that of the wild type cc4533. Fold-changes were calculated as log_10_ (C(Tx, M8)/C(Tx, wt)). C_M8_ and C_wt_ represent the contents of galactolipids in mutant08 and wild type cc4533 strains, respectively. Tx is time point. Data are presented as average ± SD (n = 3). * *p* < 0.05, ** *p* < 0.01.

**Figure 5 plants-10-00675-f005:**
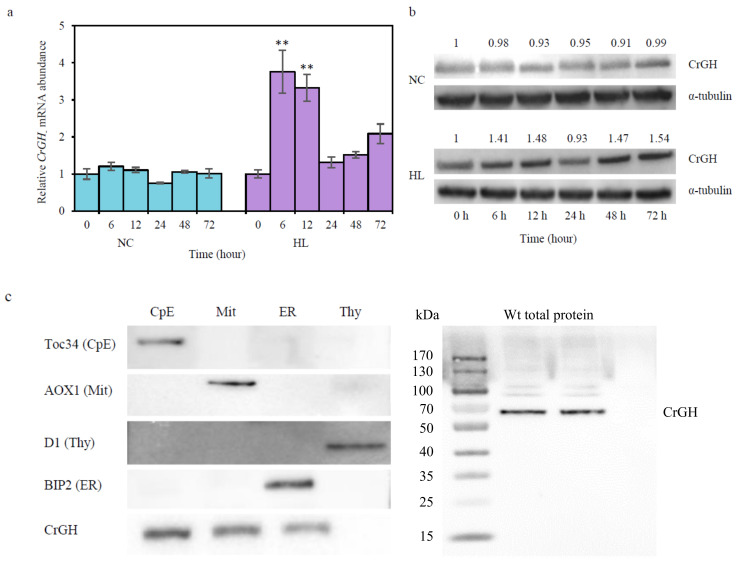
Up-regulation of *CrGH* under high light stress (HL) conditions and multi-localization of CrGH. (**a**). *CrGH* mRNA levels in wild type cc400. cells under the normal culture (NC) and high light stress conditions (HL). The abundance of mRNA was normalized to the *CrGH* mRNA level in cc400 at 0 h, which was set as 1. Data are presented as average ± SD (n = 3), which the asterisks * indicates that the *p* value of significance is less than 0.05, and ** indicates that the *p* value of significance is less than 0.01. (**b**). CrGH protein expression levels in cc400 cells, as determined by immunoblot analysis. Protein concentration was normalized to the CrGH expression level in cc400 at 0 h, which was set as 1. The α-tubulin gene was used as a reference control. (**c**). Subcellular localization of CrGH in *C. reinhardtii*. CpE, chloroplast envelope; Mit, mitochondria; ER, endoplasmic reticulum; Thy, thylakoid membrane; Wt total protein, the total protein from the wild type cc400 strain. Toc34, AOX1, D1, and BIP2 were used as markers for the CpE, Mit, Thy, and ER, respectively.

## Data Availability

All datasets generated and/or analyzed during this study are available from the corresponding author on reasonable request.

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
