# Peer review of "A Lipid Bodies-Associated Galactosyl Hydrolase Is Involved in Triacylglycerol Biosynthesis and Galactolipid Turnover in the Unicellular Green Alga Chlamydomonas reinhardtii"

_plants, 2021, doi:10.3390/plants10040675_

Round 1
Reviewer 1 Report
Gu et al have presented a manuscript describing the identification of a putative galactosyl hydrolase involved in triacylglycerol synthesis in microalga. If this study may be relevant for Plants readers, it requires to be improved before beeing considered as publishable.
Most of my concerns appear in the recombinant protein characterization :
- The authors claim that CrGH was not able to hydrolyze in vitro MGDG and DGDG, for the understanding of the manuscript, this should be presented in detail - such as the structures of lipids tested ?
- Can this absence of activity be related to the procaryotic expression of CrGH. The localization of the native enzyme in vivo in several location, post translational modification of the enzyme should be discusse din detail.
- The authors present an homology model of CrGH. No experimental data is indicated in Materials & Methods concerning the modeling of the enzyme. Which template was used, which homology ? THis model is not used by the authors, structural analysis of it may be discussed to understand why no activity was visible with the putative endogenous substrate (DGDG and MGDG)Moreover, the specificity of hydrolysis of specific acyl length observed in vivo with the knock-out studies could be discussed using this model. As such, there is no point keeping this model as it does not bring important data.
- The kinetic characterization of recombinant CrGH should be reviewed in detail. No saturation is visible for high oPNP-Gal concentration, thus the quality of the curve fitting is questionnable. higher concentration of pNP-gal should be used. Moreover, kcat value should be calculated (and not Vmax which is dependent of the enzyme concentration). The values obtained should be discussed with similar enzymes able to hydrolyze DGDG and MGDG.
Minor comments & typos:
- line 23 : "was" instead of "were"
- line 24 and line 123 : use capital D for D-galactoside.
- line 25 : The Km value is wrong (this is µM and not M)
- lines 27 and 28 : CrGH are genes or protein (is italic properly used ?)
- line 118 : simulated and not stimulated
- line 134, figure 2c : Absorbance instead of OD400. Values obtained for 500 and 100µg are not relevant, as they are >2. This should be corrected
- line 134, figure 4d : no units are given for km and Vmax. See above for providing kcat instead of Vmax
- line 363-371 : please precise the incubation volume (as no cncentration for enzyme is given). are incubations stopped with the addition of sodium carbonate ? This should be also provided...
Reviewer 2 Report
The Manuscript describes new and relevant information about the role of galactosyl hydrolase in the hydrolysis of galactolipids from photosynthetic biomembranes, and the further promotion of biosynthesis of triacylglycerols.
The experimental assays have been well designed, performed, interpreted and reported. The conclusions are supported by the experimental results.
The Manuscript is acceptable for publication with minor modifications, as detailed below.
MINOR MODIFICATIONS
Figure 2d
-Please, include vertical error bars (standard deviations) from triplicate assays around data points (mean values).
-The units of Vm and Km must be depicted.
-On the X-axis label, the concentration units of ONP-Gal must be expressed (micromolar, millimolar).
Section 5.5, Line 370
- The concentration units of ONP-Gal must be indicated (micromolar, millimolar).
Section 5.8, Line 406
-Please, indicate the name of the manufacturer and modesl of the GC-MSD equipment, as well as the type of analyser (Q, TOF, others) used for MS experiments.
Section 5.9, Line 410
-Please, indicate the name of the manufacturer and models of the HPLC-MSD equipment, as well as the type of analyzer (TOF, QTOF, QqQ, others) used for MS experiments.
Page 16
-Please correct the spelling errors in title of “Figureure 1 and 2”, replacing by “Figure 1 and 2”.
Best regards.
Reviewer 3 Report
This paper demonstrates well experimentally that galactosyl hydrolases (GrGH) of Chlamydomonas reinhardtii, a green microalga, is involved in TAG synthesis using MGDG and DGDG, the main membrane lipids of the chloroplast membrane, as substrates under hight light stress. The authors confirmed the GH enzyme activity after separating the recombinant GrGH protein by expressing it in E. coli. The authors also analyzed the differences in the content and composition of TAG and chloroplast membrane lipids under high light conditions for the Chlamydomonas mutant and wild-type knock-out of the GrGH gene. This paper revealed a new fact that GrGH is involved in TAG synthesis by decomposing DGDG in the chloroplast membrane under high light stress.
The paper is a novel result in lipid research, and the material, method, and results, discussion are well written, so it is recommended to publish.
Author Response
Thank you for your comments concerning our manuscript. Those comments are all valuable and very helpful for our paper.
Round 2
Reviewer 1 Report
All issues were addressed by the authors, I therefore consider this paper publishable in plants in its present form.
Author Response
Thank you very much for all your time and efforts for processing the manuscript.